# Integrating Coronary Plaque Information from CCTA by ML Predicts MACE in Patients with Suspected CAD

**DOI:** 10.3390/jpm12040596

**Published:** 2022-04-07

**Authors:** Guanhua Dou, Dongkai Shan, Kai Wang, Xi Wang, Zinuan Liu, Wei Zhang, Dandan Li, Bai He, Jing Jing, Sicong Wang, Yundai Chen, Junjie Yang

**Affiliations:** 1Department of Cardiology, The Second Medical Center & National Clinical Research Center for Geriatric Diseases, Chinese PLA General Hospital, Beijing 100853, China; guanhuadou@163.com; 2Department of Cardiology, Sixth Medical Center, Chinese PLA General Hospital, Beijing 100048, China; shandongkai1234@163.com (D.S.); lidandan5564@163.com (D.L.); cyundai@vip.163.com (Y.C.); 3Department of Cardiology, Yongchuan Hospital of Chongqing Medical University, Chongqing 402160, China; nkuwangkai@163.com; 4Department of Cardiology, First Medical Center, Chinese PLA General Hospital, Beijing 100853, China; plaghwangxi@163.com (X.W.); liuzinuan1995@163.com (Z.L.); zw77125@Hotmail.com (W.Z.); jsycbhhbqw@163.com (B.H.); jjing301@126.com (J.J.); 5General Electric Healthcare China, Beijing 100176, China; sicong.wang@ge.com

**Keywords:** coronary plaque, machine learning, major adverse cardiovascular events, coronary artery disease, coronary computed tomographic angiography

## Abstract

Conventional prognostic risk analysis in patients undergoing noninvasive imaging is based upon a limited selection of clinical and imaging findings, whereas machine learning (ML) algorithms include a greater number and complexity of variables. Therefore, this paper aimed to explore the predictive value of integrating coronary plaque information from coronary computed tomographic angiography (CCTA) with ML to predict major adverse cardiovascular events (MACEs) in patients with suspected coronary artery disease (CAD). Patients who underwent CCTA due to suspected coronary artery disease with a 30-month follow-up for MACEs were included. We collected demographic characteristics, cardiovascular risk factors, and information on coronary plaques by analyzing CCTA information (plaque length, plaque composition and coronary artery stenosis of 18 coronary artery segments, coronary dominance, myocardial bridge (MB), and patients with vulnerable plaque) and follow-up information (cardiac death, nonfatal myocardial infarction and unstable angina requiring hospitalization). An ML algorithm was used for survival analysis (CoxBoost). This analysis showed that chest symptoms, the stenosis severity of the proximal anterior descending branch, and the stenosis severity of the middle right coronary artery were among the top three variables in the ML model. After the 22nd month of follow-up, in the testing dataset, ML showed the largest C-index and AUC compared with Cox regression, SIS, SIS score + clinical factors, and clinical factors. The DCA of all the models showed that the net benefit of the ML model was the highest when the treatment threshold probability was between 1% and 9%. Integrating coronary plaque information from CCTA based on ML technology provides a feasible and superior method to assess prognosis in patients with suspected coronary artery disease over an approximately three-year period.

## 1. Introduction

Coronary computed tomography angiography (CCTA) is increasingly accepted as a first-line noninvasive imaging examination that has shown high accuracy for diagnosing and excluding coronary artery disease (CAD) [1,2]. Furthermore, CCTA examination was used to evaluate various stages of atherosclerosis ranging from plaque formation (length, composition, and morphology) to plaque progression, aiding in risk stratification for future major adverse cardiovascular events (MACE) and medical decision-making for patients with CAD [3,4,5,6,7].

Conventional CCTA risk scores were used to stratify the patients with CAD mainly based on the presence, length, composition, and luminal stenosis of 16-segment coronary plaque [8,9,10]. This plaque information was integrated into a single score, assuming a linear relationship between the atherosclerosis extent and outcomes [8,11,12]. Machine learning (ML) is a field of computer science that uses advanced algorithms including a great number of variables to optimize prediction, and this methodology has the potential to maximize the utilization of the coronary plaque information derived from CCTA without prior assumptions for independent variables. Previous studies have demonstrated that ML showed improves predictive values for death, myocardial ischemia and myocardial infarction compared with conventional risk scores [13,14,15]. The aim of the present study was to explore whether ML based on survival data with a time-dependent outcome integrating plaque information from CCTA exhibits better predictive values for MACEs over an approximately three-year follow-up period than the conventional CCTA risk score in patients with suspected coronary artery disease. 

## 2. Materials and Methods

### 2.1. Study Population

This is a single-center prospective observational study that was approved by the institutional review board of PLA General Hospital. All patients provided written informed consent. A total of 5526 patients with suspected coronary artery disease who sequentially underwent CCTA at the Department of Cardiology of PLA General Hospital were included from January 2015 to December 2016. The inclusion criteria were complete CCTA and clinical data. The exclusion criteria were prior known CAD (defined as prior myocardial infarction or revascularization) or those with early revascularization after CCTA (defined as within 3 months), incomplete CCTA, motion artifacts, poor-quality images, or severe coronary artery calcification that was unable to be interpreted (Figure 1). In total, 4017 patients were included.

### 2.2. Clinical Data

Demographic characteristics (age, male sex, and body mass index [BMI]) and conventional cardiovascular risk factors (dyslipidemia, hypertension, diabetes, current smoking, and family history of CAD) were collected by checking the medical record system. Hypertension was defined as a history of blood pressure >140 mmHg or treatment with antihypertensive medications. Diabetes mellitus was defined by a diagnosis made previously and/or use of insulin or oral hypoglycemic agents. Smoking was defined as current smoking or cessation of smoking within the last 3 months. A family history of premature CAD was defined as MI in a first-degree relative <55 years (male) or <65 years (female). Dyslipidemia was defined as known but untreated dyslipidemia or current treatment with lipid-lowering medications.

### 2.3. Image Acquisition and Analysis

A second-generation dual-source CT (Simens CT SOMATOM Definition Flash, SIEMENS AG, Munich, Germany) was used for the CCTA scanning. The acquisition protocols were performed in accordance with the Society of Cardiovascular Computed Tomography guidelines [16]. A detailed methodology has been previously published [17].

All images were analyzed by three radiologists or cardiologists using the 16-segment coronary artery tree model for the segment involvement score (SIS score) and the 18-segment coronary artery tree model for ML [10,16]. Plaque was defined as a tissue structure > 1 mm^2^ within or adjacent to the coronary artery lumen that could be distinguished from surrounding pericardial tissue, epicardial fat, or the vessel lumen [8]. The presence of plaque was evaluated with the corresponding stenosis severity in each segment. The coronary plaques in each segment were classified as noncalcified, mixed, and calcified plaques. The corresponding stenosis severity of the plaques was classified as 0%, 1–24%, 25–49%, 50–69%, 70–99%, and 100%. Lengths of coronary plaque were classified as 0 mm, <10 mm, 10–20 mm, and >20 mm. Coronary dominance was divided into left dominant, right dominant, and balanced types. Myocardial bridge was defined as a coronary artery segment that was surrounded by myocardium and led to systolic compression of a part of the myocardium covering the epicardial vessels [18]. Plaques with two or more characteristics (positive remodeling, spotty calcification, low attenuation plaque, and napkin-ring sign) at the same time were defined as vulnerable plaques [19]. Positive remodeling was assessed as the cross-sectional area at the site of maximal stenosis divided by an average of the proximal and distal reference segment cross-sectional areas [20]. Spotty calcification was defined by calcium deposits (>130 HU) that were <3 mm within an atheroma [21]. A low attenuation plaque was defined as a plaque with an average attenuation <30 HU, and the size of the necrotic core was >1 mm^2^ [19]. The napkin-ring sign was defined as a ring of attenuation of <130 HU that formed an arc of higher attenuation around a low attenuating plaque [22].

### 2.4. Outcome

The survival status of the patient was obtained by reviewing the electronic medical record system or patient interviews at least 90 days after CCTA examination from 1 January 2015 to 31 August 2020. MACEs, including nonfatal myocardial infarction, unstable angina requiring hospitalization, and cardiac death, were recorded as the outcome of the present study. Two physicians judged each event independently. In the case of divergence, a third physician was consulted.

### 2.5. Machine Learning Algorithm with Survival Times

Fifty-seven CCTA variables (including plaque length, plaque composition and stenosis severity of 18 coronary artery segments, coronary artery dominance, myocardial bridge, and vulnerable plaque) and nine clinical factors (including male, age, BMI, diabetes, hypertension, dyslipidemia, family history of CAD, current smoking, and chest symptoms) were available (Table 1). Machine learning involved automated feature selection, model building, and 10-fold stratified cross-validation for the entire process [23,24]. Machine learning techniques were implemented using R version 4.0.2.

First, the data were randomly divided into a training dataset and a testing dataset at a 7:3 ratio. The training dataset was used to build the prediction model, and the testing dataset was independently used to verify the effectiveness of the prediction model generated by the training dataset.

Second, automated feature selection for fifty-seven CCTA variables and nine clinical factors was performed in the training dataset using least absolute shrinkage and selection operator regression for Cox regression (LASSO-COX), which minimizes the log partial likelihood subject to the sum of the absolute values of the parameters being bounded by a constant, shrinks coefficients, and produces some coefficients that are zero, allowing for efficient variable selection (Table 1) [23].

Then, filtered CCTA variables were included for model generation. The model for MACE prediction was constructed using ‘CoxBoost’, an algorithm used to fit a Cox proportional hazards model by componentwise likelihood based on the offset-based boosting approach. This algorithm is especially suited for models with a large number of variables and allows for mandatory covariates with unpenalized parameter estimates [25,26,27,28].

The model building procedure using the training dataset included two steps, as follows. First, the hyperparameters of CoxBoost (penalty, optimal step, and numbers of estimators) were automatically calculated by the training dataset. The penalty value was calculated using a coarse line search that lead to an optimal number of boosting steps for CoxBoost, as determined by 10-fold cross-validation [29]. The optimal step of the model was confirmed using a coarse line search considering the connections between parameters to identify a potential combination of tuned hyperparameters (a penalty updating scheme was helped by an optimum step-size modification for CoxBoost), which results in an optimal model in terms of cross-validated partial log-likelihood [26]. Second, after tuning the hyperparameters from 10-fold stratified cross validation, the model was refitted on the entire training dataset for the training model. Then, the trained model was validated on the independent testing dataset (30% of entire data) to show the prediction probabilities. Compared with other models, the performance of the ML model was derived from the testing dataset.

### 2.6. The Reference Models

First, Cox proportional hazard regression (Cox regression), including the same variables as the ML model and the conventional CCTA risk score (SIS score) assessing overall plaque burden, was used in this study. The SIS score was calculated as a measure of overall coronary segments with plaque by summation of the absolute number of coronary segments with plaques (0–16) [30]. Second, the clinical factors were added to the SIS score (SIS score + clinical factors), and only clinical factors were used in this study as reference models.

### 2.7. Statistical Analysis

Continuous variables are presented as the mean ± standard deviation, and categorical variables are presented as counts (%). We assessed the performance of each prediction model (including CoxBoost, Cox regression, SIS score, SIS score + clinical factors, and clinical factors) to discriminate outcomes on the testing dataset using the C-index and AUC [31]. We evaluated the calibration of each prediction model using the Brier score [32]. The Cox regression model included the variables used in the ML model. The Brier score calculates the mean squared distance between the predicted probabilities and actual outcomes, and a smaller value indicates better calibration (<0.25 indicates significant) [32]. Decision curve analysis (DCA) of all models revealed the preferred model with the best net benefit at any given threshold. The statistical analysis was implemented in R version 4.0.2. A two-sided *p* value < 0.05 was considered statistically significant.

## 3. Results

### 3.1. Study Population

A total of 4017 patients were included in this study. The mean age was 57.76 ± 10.98 years, and 54.29% were male (Table 2). Patients without CAD, patients with nonobstructive CAD, and patients with obstructive CAD represented 37.27%, 33.06%, and 29.67% of the study population, respectively. During a mean follow-up of 29 months, 176 events (14 cardiac deaths (0.3%), 9 nonfatal myocardial infarctions (0.2%), and 190 cases of unstable angina requiring hospitalization (4.7%)) were recorded.

### 3.2. Feature Selection and Model Generation

In this study, feature selection was performed by LASSO-COX (Figure 2). When the hyperparameter of feature selection were determined (partial likelihood deviance is minimum), the algorithm output filtered variables with non-zero coefficients (chest symptoms (symptom); MB; plaque composition of the middle right coronary, the left main coronary artery, the proximal, middle and distal anterior descending branch, the first obtuse marginal branch, and the ramus intermedius artery (RCAm_composition, LM_composition, LADp_composition, LADm_composition, LADd_composition, OM1_composition, RI_composition); plaque length of the distal right coronary, the proximal anterior descending branch, and the proximal circumflex branch (RCAd_length, LADp_length, LCXp_length); and stenosis of the proximal and middle right coronary, the left main coronary artery, the proximal, middle and distal anterior descending branch, the first and second diagonal branch, and the proximal circumflex branch (RCAp_stenosis, RCAm_stenosis, LM_stenosis, LADp_stenosis, LADm_stenosis, LADd_stenosis, D1_stenosis, D2_stenosis, LCXp_stenosis)) (Figure 2).

After feature selection, the filtered variables were included in model generation (Figure 3). When the hyperparameters of the ML model were determined (the penalty was 1116, and the step was 74), the optimal model (the logplik of the 10-fold stratified cross validation was the largest) was identified in the training dataset (Figure 3a). In the ML model, chest symptoms, stenosis of the proximal anterior descending branch, and stenosis of the middle right coronary artery were among the top three variables (Figure 3b).

### 3.3. Assessment of the Performance of Each Prediction Model

After the 22nd month of follow-up, compared to other models (Cox regression, SIS score, SIS score + clinical factors, and clinical factors), the C-index of the ML model for prediction of the MACE in the testing dataset (30% of the data not used for model building) was significantly increased (C-index: 0.770–0.782, 0.723–0.752, 0.706–0.742, 0.686–0.712, 0.639–0.653, *p* < 0.05) (Figure 4 and Table 3), whereas the AUC of the ML model for the prediction of the MACE was also significantly increased in approximately three years [AUC (CI): 0.780 (0.726, 0.834), 0.738 (0.667, 0.809), 0.725 (0.669, 0.782), 0.702 (0.643, 0.762), 0.656 (0.581, 0.730), *p* < 0.05] (Figure 5 and Table 4).

### 3.4. Model Evaluation Using Calibration and DCA

In this study, we evaluated each model through calibration and DCA. In the model calibration, this study shows that the Brier score for each model to predict the MACE was less than 0.040 in approximately three years (<0.25 means significant) (Table 5). The DCA of all the models showed that the proportion of the benefit for the population each year was the highest when the risk assessment of the ML model was used for treatment, while the treatment threshold probability was between 1% and 9% over a period of approximately three years. (Figure 6).

## 4. Discussion

In this study, we used ML integrating numerous coronary plaque factors (stenosis severity, lesion length, plaque location and composition considering the 18 coronary segments, coronary dominance, myocardial bridge (MB), and patient with vulnerable plaque) and clinical and demographic information to predict MACEs after an approximately three-year period in a cohort study that accounts for time to event. The results of this study suggest that a newly generated model based on ML, accounting for nonlinearities, provided better event prediction. This study, integrating coronary plaque information from CCTA and clinical factors based on ML technology, provides a feasible and superior method to assess prognosis in patients with suspected coronary artery disease over an approximately three-year period. 

### 4.1. Risk Stratification with CCTA

Until recently, cardiac imaging studies were more inclined to use clinical and coronary plaque features (presence, extent, location, and composition) of CCTA for risk stratification of future events [33,34]. Cheruvu C showed that the maximal severity of CAD is related to major cardiovascular events [35]. The number of segments with plaque, location, and composition also improve risk assessment [36,37]. Currently, the use of CCTA information is far from insufficient, whereas the resolution of CCTA can provide massive information for mining. The conventional CCTA risk score, linear assumptions, and conventional statistical approaches may be insufficient to complete this study [38].

### 4.2. Machine Learning Algorithms Improve the Integration of Coronary Plaque Information for Survival Analysis

ML, a subset of artificial intelligence accounting for nonlinearities, is able to integrate a number of variables [11]. Cox regression is often limited for data mining purposes due to the correlation between variables, nonlinearity of variables (including potential complex interactions among them), and the possibility of overfitting.

The feasibility of ML has been demonstrated previously in CAD risk reclassification analysis. Using 25 clinical and 44 CCTA features, Motwani et al. showed that ML significantly improved the prediction of death compared with the Framinghan risk score, SSS, SIS, and the Duke prognostic index [13]. Moreover, Dey et al. showed that an ML model incorporating semiautomatically quantified measures of coronary plaque (plaque volumes, stenosis severity, lesion length, and contrast density difference) identified vessels with hemodynamically significant CAD (fractional flow reserve ≤ 0.80) with high accuracy (AUC = 0.84) [14]. Specifically, the ML model showed greater diagnostic accuracy than a conventional statistical model that utilized the exact same data. The findings above suggest that ML improves the integration of the available data for the prediction of a certain outcome.

However, these studies are similar to a cross-sectional study (as opposed to a cohort study) because the follow-up outcomes of these studies do not include survival time and only showed dichotomous outcomes (not time-dependent).

This study accounted for time to event to obtain a more appropriate risk estimation. In the ML model, chest symptoms, stenosis of the proximal anterior descending branch, and stenosis of the middle right coronary artery were among the top three factors (Figure 3), suggesting that we need to pay more attention to these characteristics in patients with suspected coronary disease. In the assessment of the model’s performance, this study shows that the ML model significantly improved the prediction of MACEs compared with other models (Cox-Boost vs. SIS score, SIS score + clinical factors, and clinical factors: C-index: 0.770–0.782, 0.706–0.742, 0.686–0.712, 0.639–0.653, *p* < 0.05; AUC (CI): 0.780 (0.726, 0.834), 0.725 (0.669, 0.782), 0.702 (0.643, 0.762), 0.656 (0.581, 0.730), *p* < 0.05) (Figure 4 and Figure 5 and Table 2 and Table 3). Specifically, the ML model showed better predicted values than a conventional statistical model (Cox regression) that utilized the exact same variables after the 22nd month of follow-up (Cox-Boost vs. Cox regression: C-index: 0.770–0.782, 0.723–0.752, *p* < 0.05; 30-month AUC (CI): 0.780 (0.726, 0.834), 0.738 (0.667, 0.809), *p* < 0.05) (Figure 4 and Figure 5 and Table 2 and Table 3).

In the model evaluation, the ML model showed great calibration for approximately three years (Brier score < 0.040), demonstrating a low difference between the predicted risk and the actual observed risk for events, and a good prediction performance (<0.25 indicates significant) (Table 5). The decision curve analysis of all models showed that the ML model was the preferred model, with the best net benefit when the treatment threshold probability was between 1% and 9% in approximately three years (Figure 6). 

This ML model can potentially translate the detailed 18-segment CCTA reads and clinical factors into an individualized risk report that might help physicians tailor preventive medical therapy. The present study established an integrated machine-learning model to predict clinical outcomes and compared it to currently available tools including SIS score, SIS score with clinical factors, and clinical factors models. The results demonstrated that the machine-learning model was feasible and easily-obtainable. Furthermore, the machine-learning model demonstrated the best performance in discrimination and calibration. The ML model could directly output MACE risk assessment within three years based on 13 non-zero variables and their coefficients in Figure 3b (symptom, LADp_stenosis, RCAm_stenosis, LCXp_length, LM_stenosis, LADm_composition, RCAp_stenosis, RI_composition, LADd_composition, OM1_composition, LCXp_stenosis, RCAd_length). For individualized preventive therapy, as is shown in present study, the proportion of the benefit for the population each year was between 0% and 3% when the risk assessment of the ML model was used for treatment, while the treatment threshold probability was between 1% and 9% over a period of approximately three years (Figure 6). Considering the incidence of MACE events (4.4%), the proportion of the benefit for the population each year of 3% is relatively better.

### 4.3. Study Limitations

This study, which was designed as a respective single-center cohort study, was performed in a middle-aged population with suspected coronary artery disease. Therefore, the results of this study may not be generalizable to other study populations. This study was lacking in medication history and only followed up after nearly three years. Further research may follow up for longer, add follow-up medication history, include genetic data, and identify the image feature-genome interaction, wihle combined prediction ability may potentially improve the risk estimation.

## 5. Conclusions

Integrating coronary plaque information from CCTA based on machine learning technology provides a feasible and superior method to assess prognosis in patients with suspected coronary artery disease over an approximately three-year period.

## Figures and Tables

**Figure 1 jpm-12-00596-f001:**
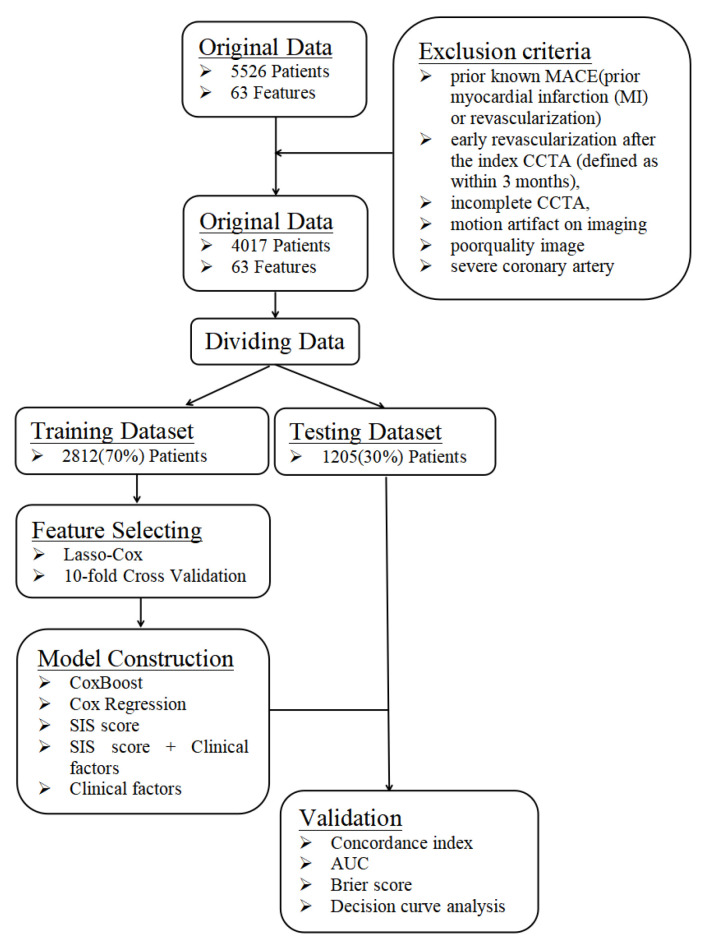
A flowchart about the framework of this study. The data were randomly divided into a training dataset and a testing dataset at a ratio of 7:3. The training dataset was used to build the prediction model, whereas the testing dataset was independently used to verify the effectiveness of the prediction model generated by the training dataset by computing C-index, AUC, Brier score and DCA.

**Figure 2 jpm-12-00596-f002:**
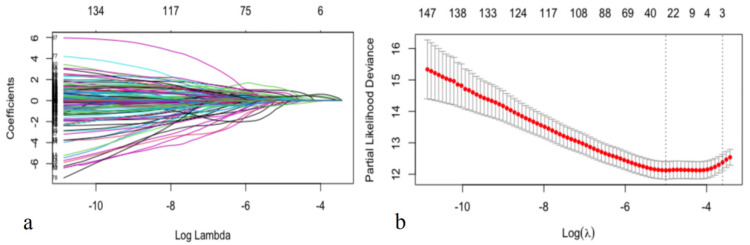
Selecting process for features by Lasso-Cox. Automated feature selection for fifty-seven CCTA variables and nine clinical factors was performed using LASSO-COX, which minimizes the log partial likelihood subject to the sum of the absolute values of the parameters being bounded by a constant, shrinks coefficients, and produces some coefficients that are zero, allowing efficient variable selection (**a**). When the hyperparameters of feature selection were determined (partial likelihood deviance is minimum) (**b**), the algorithm outputted 21 filtered variables with non-zero coefficients (the filtered variables were included in model generation subsequently).

**Figure 3 jpm-12-00596-f003:**
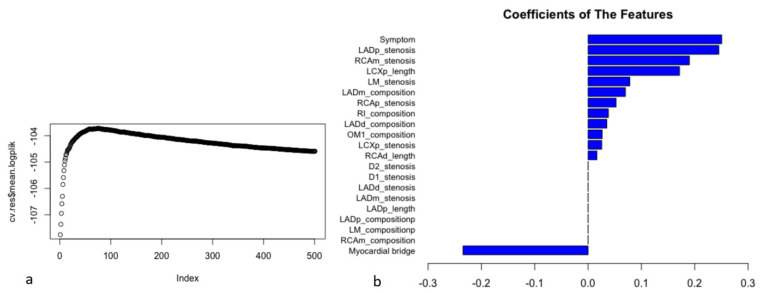
The process of model construction and coefficients of the features in the ML model. Filtered CCTA variables were included in model generation. The 21 filtered variables with non-zero coefficients in the results of LASSO-COX were included in ML model generation. The hyperparameters of ML model were automatically calculated on the training dataset. After tuning the hyperparameters (the penalty was 1116, and the step was 74) from 10-fold stratified cross validation, the model was refitted on the entire training dataset for training model. When the logplik of the 10-fold stratified cross validation was the largest (cv.res$mean.logplik = −103.723), it showed the optimal model in the training dataset and the coefficients of features (**a**). In the ML model, chest symptoms (symptom), the stenosis severity of the proximal anterior descending branch (LADp_stenosis), and the stenosis severity of the middle right coronary artery (RCAm_stenosis) were among the top three variables (coefficients: 0.251, 0.245, and 0.190, respectively) (**b**).

**Figure 4 jpm-12-00596-f004:**
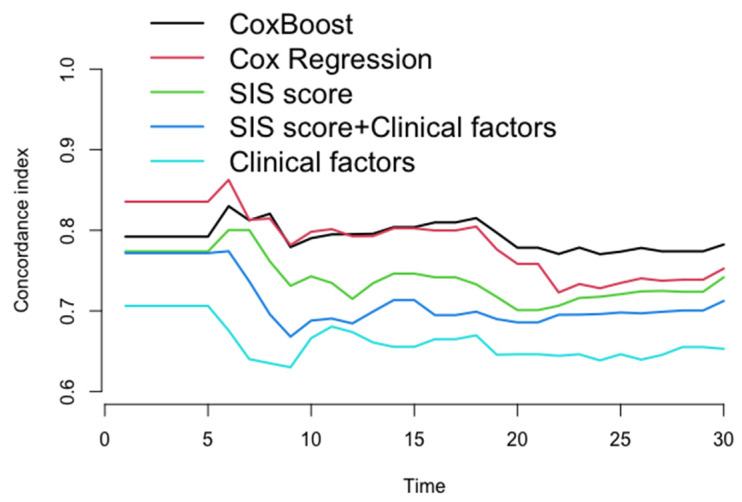
The concordance index for each model in testing dataset every month. After 22nd month in follow-up, compared to other models (Cox regression, SIS score, SIS score + clinical factors, and clinical factors), the C-index of ML model for prediction of the MACE in the testing dataset was significantly increased (C-index: 0.770–0.782, 0.723–0.752, 0.706–0.742, 0.786–0.712, 0.639–0.653, *p* < 0.05).

**Figure 5 jpm-12-00596-f005:**
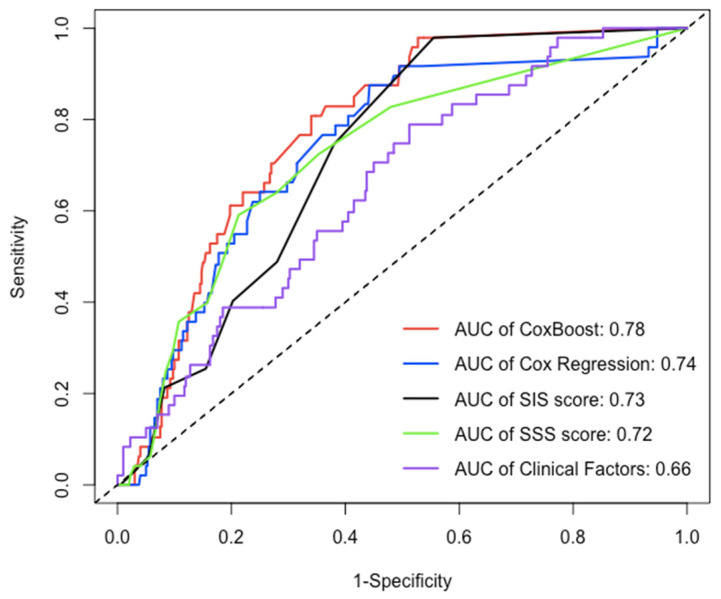
The AUC of each model in testing dataset over 30 months. Over an approximately three-year period, compared to the AUC of other models (Cox regression, SIS score, SIS score + clinical factors, and clinical factors), the AUC of ML model for prediction of MACE was significantly increased [AUC(CI): 0.780 (0.726, 0.834), 0.738 (0.667, 0.809), 0.725 (0.669, 0.782), 0.702 (0.643, 0.762), 0.656 (0.581, 0.730), *p* < 0.05].

**Figure 6 jpm-12-00596-f006:**
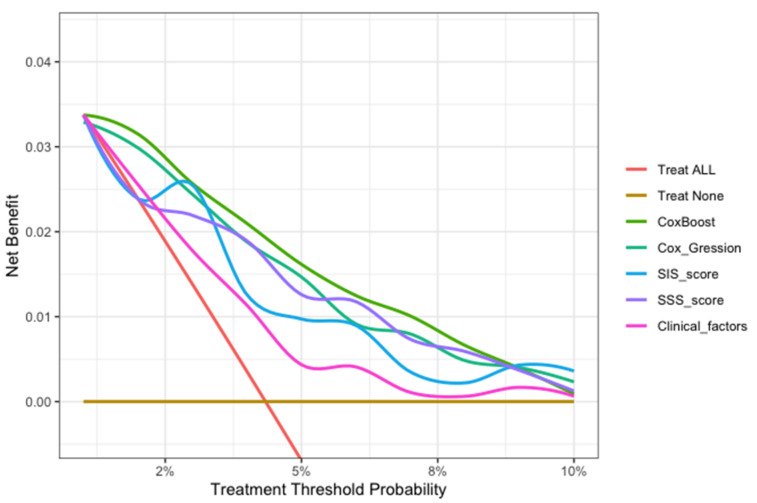
The decision curve analysis of all models for patients over 30 months. The brown transverse line = net benefit when all patients are considered as not having the outcome (MACEs); red oblique line = net benefit when all patients are considered as having the outcome (MACEs). The decision curve analysis of all models showed that the proportion of the benefit for the population each year was the highest when the risk assessment of the ML model was used for treatment, while the treatment threshold probability was between 1% and 9% over a period of approximately three years.

**Table 1 jpm-12-00596-t001:** Features Selected by Lasso-Cox.

Features	Definition	Category
Demographic characteristics		
Age	Age of the patient	continuous variable
BMI	Body mass index	continuous variable
Male	Are they male?	1/0 = yes/no
Cardiovascular risk factors		
Symptom	Types of chest pain	0/1/2 = no/atypical/typical
Hyperlipemia	Is there hyperlipemia	1/0 = yes/no
Hypertension	Is there hypertension	1/0 = yes/no
Diabetes	Is there diabetes	1/0 = yes/no
Currently smoking	Are they currently smoking	1/0 = yes/no
Family history of CAD	Is there family history for CAD	1/0 = yes/no
CCTA Features		
Coronary dominance	Is there left/right/balanced dominance?	1/2/3 = left/right/balanced
Myocardial bridge	Is there myocardial bridge?	1/0 = yes/no
Vulnerable plaque	Are there two or more characteristics of vulnerable plaque?	1/0 = yes/no
RCAp_composition	Composition of plaque in proximal RCA	0/1/2/3 = normal/calcified/non-calcified/mix
RCAm_composition	Composition of plaque in middle RCA	0/1/2/3 = normal/calcified/non-calcified/mix
RCAd_composition	Composition of plaque in distal RCA	0/1/2/3 = normal/calcified/non-calcified/mix
P-PDA_composition	Composition of plaque in PDA of RCA origin	0/1/2/3 = normal/calcified/non-calcified/mix
LM_composition	Composition of plaque in LM	0/1/2/3 = normal/calcified/non-calcified/mix
LADp_composition	Composition of plaque in proximal LAD	0/1/2/3 = normal/calcified/non-calcified/mix
LADm_composition	Composition of plaque in middle LAD	0/1/2/3 = normal/calcified/non-calcified/mix
LADd_composition	Composition of plaque in distal LAD	0/1/2/3 = normal/calcified/non-calcified/mix
D1_composition	Composition of plaque in D1	0/1/2/3 = normal/calcified/non-calcified/mix
D2_composition	Composition of plaque in D2	0/1/2/3 = normal/calcified/non-calcified/mix
LCXp_composition	Composition of plaque in proximal LCX	0/1/2/3 = normal/calcified/non-calcified/mix
OM1_composition	Composition of plaque in OM1	0/1/2/3 = normal/calcified/non-calcified/mix
LCXd_composition	Composition of plaque in distal LCX	0/1/2/3 = normal/calcified/non-calcified/mix
OM2_composition	Composition of plaque in OM2	0/1/2/3 = normal/calcified/non-calcified/mix
L-PDA_composition	Composition of plaque in PDA of LAD origin	0/1/2/3 = normal/calcified/non-calcified/mix
R-PLB_composition	Composition of plaque in PLB of RCA origin	0/1/2/3 = normal/calcified/non-calcified/mix
RI_composition	Composition of plaque in RI	0/1/2/3 = normal/calcified/non-calcified/mix
L-PLB_composition	Composition of plaque in PLB of LAD origin	0/1/2/3 = normal/calcified/non-calcified/mix
RCAp_length	Length of plaque in proximal RCA	0/1/2/3 = normal/localized/segmental/diffuse
RCAm_length	Length of plaque in middle RCA	0/1/2/3 = normal/localized/segmental/diffuse
RCAd_length	Length of plaque in distal RCA	0/1/2/3 = normal/localized/segmental/diffuse
P-PDA_length	Length of plaque in PDA of RCA origin	0/1/2/3 = normal/localized/segmental/diffuse
LM_length	Length of plaque in LM	0/1/2/3 = normal/localized/segmental/diffuse
LADp_length	Length of plaque in proximal LAD	0/1/2/3 = normal/localized/segmental/diffuse
LADm_length	Length of plaque in middle LAD	0/1/2/3 = normal/localized/segmental/diffuse
LADd_length	Length of plaque in distal LAD	0/1/2/3 = normal/localized/segmental/diffuse
D1_length	Length of plaque in D1	0/1/2/3 = normal/localized/segmental/diffuse
D2_length	Length of plaque in D2	0/1/2/3 = normal/localized/segmental/diffuse
LCXp_length	Length of plaque in proximal LCX	0/1/2/3 = normal/localized/segmental/diffuse
OM1_length	Length of plaque in OM1	0/1/2/3 = normal/localized/segmental/diffuse
LCXd_length	Length of plaque in distal LCX	0/1/2/3 = normal/localized/segmental/diffuse
OM2_length	Length of plaque in OM2	0/1/2/3 = normal/localized/segmental/diffuse
L-PDA_length	Length of plaque in PDA of LAD origin	0/1/2/3 = normal/localized/segmental/diffuse
R-PLB_length	Length of plaque in PLB of RCA origin	0/1/2/3 = normal/localized/segmental/diffuse
RI_length	Length of plaque in RI	0/1/2/3 = normal/localized/segmental/diffuse
L-PLB_length	Length of plaque in PLB of LAD origin	0/1/2/3 = normal/localized/segmental/diffuse
RCAp_stenosis	Stenosis of plaque in proximal RCA	0/1/2/3/4 = normal/mininal/mild/moderate/severe
RCAm_stenosis	Stenosis of plaque in middle RCA	0/1/2/3/4 = normal/mininal/mild/moderate/severe
RCAd_stenosis	Stenosis of plaque in distal RCA	0/1/2/3/4 = normal/mininal/mild/moderate/severe
P-PDA_stenosis	Stenosis of plaque in PDA of RCA origin	0/1/2/3/4 = normal/mininal/mild/moderate/severe
LM_stenosis	Stenosis of plaque in LM	0/1/2/3/4 = normal/mininal/mild/moderate/severe
LADp_stenosis	Stenosis of plaque in proximal LAD	0/1/2/3/4 = normal/mininal/mild/moderate/severe
LADm_stenosis	Stenosis of plaque in middle LAD	0/1/2/3/4 = normal/mininal/mild/moderate/severe
LADd_stenosis	Stenosis of plaque in distal LAD	0/1/2/3/4 = normal/mininal/mild/moderate/severe
D1_stenosis	Stenosis of plaque in D1	0/1/2/3/4 = normal/mininal/mild/moderate/severe
D2_stenosis	Stenosis of plaque in D2	0/1/2/3/4 = normal/mininal/mild/moderate/severe
LCXp_stenosis	Stenosis of plaque in proximal LCX	0/1/2/3/4 = normal/mininal/mild/moderate/severe
OM1_stenosis	Stenosis of plaque in OM1	0/1/2/3/4 = normal/mininal/mild/moderate/severe
LCXd_stenosis	Stenosis of plaque in distal LCX	0/1/2/3/4 = normal/mininal/mild/moderate/severe
OM2_stenosis	Stenosis of plaque in OM2	0/1/2/3/4 = normal/mininal/mild/moderate/severe
L-PDA_stenosis	Stenosis of plaque in PDA of LCX origin	0/1/2/3/4 = normal/mininal/mild/moderate/severe
R-PLB_stenosis	Stenosis of plaque in PLB of RCA origin	0/1/2/3/4 = normal/mininal/mild/moderate/severe
RI_stenosis	Stenosis of plaque in RI	0/1/2/3/4 = normal/mininal/mild/moderate/severe
L-PLB_stenosis	Stenosis of plaque in PLB of LCX origin	0/1/2/3/4 = normal/mininal/mild/moderate/severe

BMI, body mass index; CAD, coronary artery disease; CCTA, coronary computed tomography angiography; RCA, right coronary artery; PDA, posterior descending artery; LM, left main coronary artery; LAD, left anterior descending branch; D1, first diagonal branches; D2, second diagonal branches; LCX, left circumflex branch; OM1, first obtuse marginal branch; OM2, second obtuse marginal branch; PLB, posterior lateral branch; RI, intermediate ramus.

**Table 2 jpm-12-00596-t002:** Demographic and Clinical Characteristics of Patients at Baseline.

Characteristics	Total (*n* = 4017)	Training Dataset (*n* = 2812)	Testing Dataset (*n* = 1205)
Age (y)	57.76 ± 10.98	57.43 ± 10.94	57.71 ± 10.86
Male (*n*, %)	2181 (54.29)	1544 (54.91)	637 (52.86)
BMI (kg/m^2^)	25.47 ± 3.41	25.50 ± 3.43	25.40 ± 3.34
SIS score	1.80 ± 4.17	1.82 ± 2.05	1.74 ± 2.03
Follow-up time (months)	29.56 ± 5.94	29.51 ± 6.09	29.68 ± 5.57
Chest symptom			
No chest pain (*n*, %)	1935 (48.17)	1338 (47.58)	597 (49.54)
Atypical chest pain (*n*, %)	1692 (42.12)	1192 (42.39)	500 (41.49)
Typical chest pain (*n*, %)	390 (9.71)	282 (10.03)	108 (8.96)
Cardiovascular risk factors			
Hyperlipemia (*n*, %)	1311 (32.64)	912 (32.43)	399 (33.11)
Hypertension (*n*, %)	1916 (47.70)	1333 (47.40)	583 (48.38)
Diabetes (*n*, %)	660 (16.43)	451 (16.04)	209 (17.34)
Currently smoking (*n*, %)	1023 (25.47)	716 (25.46)	307 (25.48)
Family history of CAD (*n*, %)	845 (21.04)	593 (21.09)	252 (20.91)
CCTA Finding			
No CAD (*n*, %)	1497 (37.27)	1029 (36.6)	468 (38.8)
Non-obstructive CAD (*n*, %)	1328 (33.06)	917 (32.6)	411 (34.1)
Obstructive CAD (*n*, %)	1192 (29.67)	866 (30.8)	326 (27.1)
Vulnerable plaque (*n*, %)	35 (0.87)	24 (0.85)	11 (0.91)
Myocardial bridge (*n*, %)	332 (8.26)	221 (7.86)	111 (9.21)
Coronary dominance			
Left dominant (*n*, %)	3736 (93.00)	2613 (92.92)	1123 (93.20)
Right dominant (*n*, %)	198 (4.93)	138 (4.91)	60 (4.98)
Balanced type (*n*, %)	83 (2.07)	61 (2.17)	22 (1.83)

Values are means ± SD or counts (%). BMI, body mass index; CAD, coronary artery disease; CCTA, coronary computed tomography angiography.

**Table 3 jpm-12-00596-t003:** The performance (concordance-index) for each model validated at each half year of follow-up.

Model	6th Month C-Index	12th Month C-Index	18th Month C-Index	24th Month C-Index	30th Month C-Index
CoxBoost	83.0	79.5	81.5	77.0	78.2
Cox regression	86.3	79.3	80.5	72.8	75.2
SIS score	80.0	71.5	73.3	71.8	74.2
SIS score + clinical factors	77.4	68.4	69.9	69.6	71.2
Clinical factors	67.6	67.4	67.0	63.9	65.3

Cox regression, Cox proportional hazard regression; SIS score, segment involvement score; SIS score + clinical factors, clinical factors added to segment involvement score.

**Table 4 jpm-12-00596-t004:** Comparison of AUC for each model validated at 30 months of follow-up.

Model	AUC	95%CI	*p* (CoxBoost vs.)
CoxBoost	0.780	0.726, 0.834	\
Cox regression	0.738	0.667, 0.809	0.048
SIS score	0.725	0.669, 0.782	0.010
SIS score + clinical factors	0.702	0.643, 0.762	0.003
Clinical factors	0.656	0.581, 0.730	0.005

AUC, area under the receiver operator characteristic curve; Cox regression, Cox proportional hazard regression; SIS score, segment involvement score; SIS score + clinical factors, clinical factors added to segment involvement score.

**Table 5 jpm-12-00596-t005:** The calibration (Brier score) for each model validated at each half year of follow-up.

Model	6th Month BS	12th Month BS	18th Month BS	24th Month BS	30th Month BS
CoxBoost	0.004	0.006	0.020	0.033	0.039
Cox regression	0.004	0.012	0.021	0.033	0.039
SIS score	0.006	0.012	0.021	0.033	0.039
SIS score + clinical factors	0.004	0.011	0.020	0.033	0.039
Clinical factors	0.004	0.011	0.020	0.033	0.039

BS, Brier score; Cox regression, Cox proportional hazard regression; SIS score, segment involvement score; SIS score + clinical factors, clinical factors added to segment involvement score.

## Data Availability

Not applicable.

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
