# Peer review of "Integrating Coronary Plaque Information from CCTA by ML Predicts MACE in Patients with Suspected CAD"

_jpm, 2022, doi:10.3390/jpm12040596_

Round 1

Reviewer 1 Report

In this paper, the authors have merged coronary plaque information from CCTA by machine learning predicts MACE in patients with suspected CAD. This paper is written well. I have a few comments.

  1. Please highlight the contributions of this paper in the introduction section.
  2. The advantages and the disadvantages of your work must be written.

Author Response

Point 1: Please highlight the contributions of this paper in the introduction section. 

 Response 1: Thanks to the reviewer’s comment. The aim of the present study was to explore whether an ML method based on survival data(time-dependent outcome) integrating plaque information from CCTA exhibits better predictive value for MACEs over an approximately three-year follow-up period than the conventional CCTA risk score in patients with suspected coronary artery disease. It has been corrected in the introduction section.

Point 2: The advantages and the disadvantages of your work must be written.

Response 2: Thanks to the reviewer’s comment. 1)Advantages: This study, integrating coronary plaque information from CCTA and clinical factors based on ML technology, provides a feasible and superior method to assess prognosis in patients with suspected coronary artery disease over an approximately three-year period. It has been corrected in added in the study discussion.

2)Disadvantages: This study, which was designed as a respective single-center cohort study, was performed in a middle-aged population with suspected coronary artery disease. Therefore, the results of this study may not be generalizable to other study populations. This study only underwent follow-up nearly three years and lack of medication history. Future research may follow longer, add follow-up medication history, include genetic data, identify the image feature-genome interaction and combined prediction ability may potentially improve the risk estimation. It has been added in the study limitation .

Reviewer 2 Report

This study builds a model using a combination of image features and clinical features. The results show that integrating the image features can effectively improve the prediction performance. 

One concern is the contribution of this manuscript is not strong. Machine learning methods combining images features and clinical features have been successfully applied in existing work. For the survival analysis, the key finding is that coxboost achieved a better performance than cox regression after 24 months. More discussion/analysis on this finding should be helpful.

Some questions in the method design:

  1. Which dataset was used for the feature selection? 
  2. Were those features normalized/standardized?

Some comments in the presentation:

  1. It seems the “validation set” and “test set” are used to refer to the same dataset in this manuscript. It is confusing.
  2. In this manuscript, ‘ML algorithm’ and ‘’ML method’ are used to refer to feature selection + coxBoost. I would suggest using coxBoost directly. Machine learning is a broad field. Methods from machine learning can be very different.
  3. Line 299-300: “ML, a subset of artificial intelligence without problems of nonconvergence and over fitting while accounting for nonlinearities” It is not true
  4. Line 236: (30% of the ML model not used for model building). A typo? “30% of the ML model”
  5. Feature extraction in 2.3. is not easy to follow. I would suggest using a table to list different categories of features, the definition of the features, and the number of features in each category.

Author Response

Point 1: Which dataset was used for the feature selection? 

Response 1: Thanks to the reviewer’s comment. Automated feature selection for fifty-seven CCTA variables and night clinical factors was performed in the training dataset using least absolute shrinkage and selection operator regression for Cox regression (LASSO-COX). It has been corrected in lines 138-140 on pages 4.

Point 2: Were those features normalized/standardized?

Response 2: With the exception of age and BMI, these features were dichotomous or categorical variables which did not need to be normalized/standardized.  

Point 3: It seems the “validation set” and “test set” are used to refer to the same dataset in this manuscript. It is confusing.

Response 3: Thanks to the reviewers’ comment. There were only two kinds of dataset in this study, which were training set and test set. Test set was used for validation of the ML prediction model. We have corrected the related content both in the manuscript and in the flow chart.

Point 4: In this manuscript, ‘ML algorithm’ and ‘’ML method’ are used to refer to feature selection + coxBoost. I would suggest using coxBoost directly. Machine learning is a broad field. Methods from machine learning can be very different.

Response 4: Thanks to the reviewers’ comment.It has been corrected in the manuscript.

Point 5: Line 299-300: “ML, a subset of artificial intelligence without problems of nonconvergence and over fitting while accounting for nonlinearities” It is not true

Response 5: Thanks for the suggestion. ML, a subset of artificial intelligence accounting for nonlinearities, is able to integrate a number of variables. It has been corrected in lines 317-318 on pages 11 in the study discussion.

Point 6: Line 236: (30% of the ML model not used for model building). A typo? “30% of the ML model”

Response 6: Thanks for the suggestion. “The C-index of ML model for prediction of the MACE in the testing dataset (30% of the data not used for model building) was significantly increased”. It has been corrected in lines 246-248 on pages 8 in the study results.

Point 7: Feature extraction in 2.3. is not easy to follow. I would suggest using a table to list different categories of features, the definition of the features, and the number of features in each category.

Response 7: Thanks for the suggestion. Automated feature selection for fifty-seven CCTA variables and night clinical factors was performed in the training dataset using least absolute shrinkage and selection operator regression for Cox regression (LASSO-COX)(Table 1). The fifty-seven CCTA variables and night clinical factors have been showing in the table 1 on page 4-6.

Reviewer 3 Report

The paper develops a model for time-to-event data, using recently devised methods from the ML literature in the context of hazard function regression. The authors motivate the subject and give appropriate refrerences. Their methodology is compared to the standard Cox proportional hazards regression (CoxReg), and a few other methods suitable for this setting.

I have some comments and suggestions.

1) In sec 3.2, the authors start with a group of 57 covariates and then "shrink" this list, in essence doing feature selection via a sparsity seeking algorithm. The resulting short list of covariates are then referred to in list 2(b), but I don't see this list anywhere... (Also mention how many covariates are in this list.)   

2) One of the competing methods that the methodology (CoxBoost) is compared to is CoxReg. I presume that CoxReg was based on the same short list of covariates that were used for CoxBoost?

3) Sec 3.4: decision curve analysis (DCA) is used, but never explained. In fact, the reader only finds this spelled out in Fig 6. Please briefly explain what this is and give a ref.

4) Finally, there are a few grammatically incorrect statements and typographical errors.

Author Response

Point 1: In sec 3.2, the authors start with a group of 57 covariates and then "shrink" this list, in essence doing feature selection via a sparsity seeking algorithm. The resulting short list of covariates are then referred to in list 2(b), but I don't see this list anywhere... (Also mention how many covariates are in this list.)

 Response 1: Thanks to the reviewer’s comment. When the hyper-parameter of feature selection were determined (partial likelihood deviance is minimum)(2b), the algorithm outputted 21 filtered variables with non-zero coefficients(the filtered variables were included in model generation subsequently(3b)). It has been corrected in lines 222-224 on pages 7.

Point 2: One of the competing methods that the methodology (CoxBoost) is compared to is CoxReg. I presume that CoxReg was based on the same short list of covariates that were used for CoxBoost?

Response 2: Thanks to the reviewer’s comment. CoxReg was based on the same short list of covariates that were used for CoxBoost(filtered variables with non-zero coefficients in the results of LASSO, showing in the Feature selection and model generation of the manuscript). However, comparing to the CoxReg, CoxBoost is an algorithm used to fit a Cox proportional hazards model by componentwise likelihood based on the offset-based boosting approach. This algorithm is especially suited for models with a large number of variables and allows for mandatory covariates with unpenalized parameter estimates(lines 145-149 on pages 4).

Point 3: Sec 3.4: decision curve analysis (DCA) is used, but never explained. In fact, the reader only finds this spelled out in Fig 6. Please briefly explain what this is and give a ref.

Response 3: Thanks to the reviewer’s comment. The DCA of all models in Fig 6 showed that when the treatment threshold probability was between 1% to 9%, compared with other models, the proportion of the benefit population each years was the highest when the risk assessment of ML was used for treatment in approximately three years. It has been corrected in lines 277-281 on pages 10. 

Point 4: Finally, there are a few grammatically incorrect statements and typographical errors.

Response 4: Thanks for the suggestion. It has been corrected in lines 19 on page 1 and lines 129 on pages 4.

Reviewer 4 Report

This is a single-center prospective study that analysed the predictive value of integrating coronary plaque information from computed tomographic angiography (CCTA) and machine learning (ML) to predict major adverse cardiovascular events (MACEs) in patients with suspect coronary artery disease. Consecutive 4017 patients were included in the prediction model analysis. The authors noted feasibility of this method to assess prognosis over 3 years period.

Major Comments:

  1. Sample size: large sample size but single-centre as highlighted by author.
  2. Methods: Comprehensive approach. Second generation dual-source CT utilised in the study with images were analysed by 3 radiologists or cardiologists. Relevant categorisation of coronary artery features, including vulnerable plaque features. MACEs were assessed by 2 physicians independently, or third in the case of divergence.
  3. Results: I am surprised that despite such a large datasets available, only very small proportion with vulnerable plaque. Were these factors specifically analysed for? The events rate was relatively low, despite relatively well spread of normal vs non-obstructive vs obstructive. It would be interesting to know about the background therapy over the follow-up period, comparing those on guideline recommended treatment goal and without (i.e statin or ACEi/ARB or antiplatelet therapy).
  4. Discussion: How easy can this be integrated into clinical practice in comparison with currently available tools for individualised preventive therapy?

Minor Comments:

  1. Abstract: (Line 19) – … Therefore, this paper …

Author Response

Point 1: Sample size: large sample size but single-centre as highlighted by author.

Response 1: Thanks to the reviewer’s comment. 

Point 2: Methods: Comprehensive approach. Second generation dual-source CT utilised in the study with images were analysed by 3 radiologists or cardiologists. Relevant categorisation of coronary artery features, including vulnerable plaque features. MACEs were assessed by 2 physicians independently, or third in the case of divergence.

Response 2: Thanks to the reviewer’s comment.

Point 3: I am surprised that despite such a large datasets available, only very small proportion with vulnerable plaque. Were these factors specifically analysed for? The events rate was relatively low, despite relatively well spread of normal vs non-obstructive vs obstructive. It would be interesting to know about the background therapy over the follow-up period, comparing those on guideline recommended treatment goal and without (i.e statin or ACEi/ARB or antiplatelet therapy).

Response 3:.Thanks for the comment. The small proportion with vulnerable plaque is partly due to the different definition of the vulnerable plaque. In the present study, vulnerable plaque was defined as plaque with two or more characteristics simultaneously, which accords with published guideline. The related characteristics include positive remodeling, spotty calcification, low attenuation plaque, and napkin-ring sign. However, Previous studies often referred the vulnerable plaque to as one of any above mentioned characteristics. Another explanation for the small proportion with vulnerable plaque is that the included population was at relative low risk, which was in line with the appropriate use of cardiac CT scan. We admitted that the lack of medication background was a major limitation in the present study, which was mentioned in the limitation section of the manuscript (in page 14 line 389) as following:

  “This study was lack of medication history and only followed up nearly three years. Further research may follow longer, add follow-up medication history, include genetic data, identify the image feature-genome interaction and combined prediction ability may potentially improve the risk estimation.

Point 4: How easy can this be integrated into clinical practice in comparison with currently available tools for individualised preventive therapy?

Response 4: Thanks for the comment. The present study established an integrated Machine-learning model to predict clinical outcome and compared with currently available tools including SIS score, SIS score with Clinical factors, and Clinical factors models. The result demonstrated that Machine-learning model was feasible and easy-obtainable. Furthermore, Machine-learning model showed the best performance in discrimination and calibration. The ML model could directly output MACE risk assessment within three years based on 13 non-zero variables and its coefficients (Symptom, LADp_stenosis, RCAm_stenosis, LCXp_length, LM_stenosis, LADm_composition, RCAp_stenosis, RI_composition, LADd_composition, OM1_composition, LCXp_stenosis, RCAd_length). For individualised preventive therapy, as is shown in present study, when the treatment threshold probability was between 1% to 9%, integrated Machine-learning model could result in the proportion of the benefit population each years by 3% in compared to currently available tools. Considering the low incidence of MACE events (4.4%), the highest annual benefit population of 3% is already relatively high. It has been corrected in lines 355-368 on pages 12.

Point 5: Abstract: (Line 19) – … Therefore, this paper …

Response 5: Thanks for the suggestion. Therefore, this paper aimed to explore the predictive value of integrating coronary plaque information from coronary computed tomographic angiography (CCTA) with ML to predict major adverse cardiovascular events (MACEs) in patients with suspected coronary artery disease(CAD). It has been corrected in lines 19 on page 1.

Round 2

Reviewer 2 Report

My comments have been well addressed.